# Flexural Strengthening of Concrete Slab-Type Elements with Textile Reinforced Concrete

**DOI:** 10.3390/ma13102246

**Published:** 2020-05-13

**Authors:** Hyeong-Yeol Kim, Young-Jun You, Gum-Sung Ryu, Kyung-Taek Koh, Gi-Hong Ahn, Se-Hoon Kang

**Affiliations:** Structural Engineering Department, Korea Institute of Civil Engineering and Building Technology (KICT), 283 Goyangdae-Ro, Ilsanseo-Gu, Goyang, Gyeonggi-Do 10223, Korea; hykim1@kict.re.kr (H.-Y.K.); ryu0505@kict.re.kr (G.-S.R.); ktgo@kict.re.kr (K.-T.K.); agh0530@kict.re.kr (G.-H.A.); kshun@kict.re.kr (S.-H.K.)

**Keywords:** flexural strengthening, mechanical testing, fabric reinforced cementitious matrix (FRCM), structural testing, textile reinforced concrete (TRC)

## Abstract

This paper deals with flexural strengthening of reinforced concrete (RC) slabs with a carbon textile reinforced concrete (TRC) system. The surface coating treatment was applied to a carbon grid-type textile to increase the bond strength. Short fibers were incorporated into the matrix to mitigate the formation of shrinkage-induced cracks. The tensile properties of the TRC system were evaluated by a direct tensile test with a dumbbell-type grip method. The tensile test results indicated that the effect of the surface coating treatment of the textile on the bonding behavior of the textile within the TRC system was significant. Furthermore, the incorporation of short fibers in the matrix was effective to mitigate shrinkage-induced crack formation and to improve the tensile properties of the TRC system. Six full-scale slab specimens were strengthened with the TRC system and, subsequently, failure tested. The ultimate load-carrying capacity of the strengthened slabs was compared with that of an unstrengthened slab as well as the theoretical solutions. The failure test results indicated that the stiffness and the ultimate flexural capacity of the strengthened slab were at least 112% and 165% greater, respectively, than that of the unstrengthened slab. The test results further indicated that the strengthening effect was not linearly proportional to the amount of textile reinforcement.

## 1. Introduction

In the past two decades, extensive studies on the flexural strengthening of concrete structures with textile reinforced concrete (TRC) have been carried out. TRC consists of textile reinforcement and a cementitious matrix and, in the literature, is also known as fabric-reinforced cementitious matrix (FRCM). Early developments of TRC and recent applications to the flexural strengthening of concrete structures are well summarized in [1,2,3,4,5].

The effectiveness of flexural strengthening of reinforced concrete (RC) beams with carbon TRC systems has been studied experimentally by several research groups [6,7,8,9]. Flexural strengthening of one-way RC slabs [10,11], as well as two-way RC slabs [12], with TRC has also been investigated. In their studies, fiber type, number of textile plies, and matrix strength were generally considered as design factors affecting TRC strengthening. Overall, the flexural capacity of an RC member increases as the number of textile plies is increased, but this gain in flexural capacity is not linearly proportional to the amount of textile reinforcement. The use of fibers with a higher elastic tensile modulus such as carbon fibers is more effective in terms of flexural strengthening than using those with a lower value such as glass fibers. The influence of matrix strength on the TRC strengthening effect is generally insignificant. The results of the existing studies indicate that, if properly designed and installed, the TRC system can effectively be used for flexural strengthening of RC members.

The bonding strength of carbon textile is generally smaller than that of steel rebars. The influence of surface treatment of textiles on the bond properties has been investigated by several research groups [13,14,15,16,17]. The results of their experimental works have confirmed that the surface treatment of textile is an effective method to improve the bond strength of the textile.

Meanwhile, the influence of short fibers on the mechanical properties of TRC has been investigated by numerous research groups [18,19,20,21]. The results of their studies indicate that the presence of short fibers within the cementitious matrix can improve the bonding between the textile and matrix, and thereby the flexural, tensile, and cracking strength of TRC can be improved [3,18]. Mansouri et al. [22] studied the influence of steel fibers on the abrasive resistance of concrete. The results indicated that the abrasive resistance of concrete can also be enhanced by incorporating steel fibers in concrete. Furthermore, the curing conditions significantly affect the material properties of the cementitious matrix, grout, and concrete. The influence of temperature and humidity on the behavior of grouts [23] and the compressive strength of concrete has also been studied [24].

This paper presents an experimental validation of RC slab-type elements strengthened with a carbon TRC system. The surface of a carbon grid-type textile was coated with an abrasive powder material to increase the bond strength. The matrix was reinforced with short fibers to mitigate shrinkage-induced crack formation. The surface coating of the textile, short fiber volume fractions, and number of textile plies were considered as design parameters to design the TRC system.

The tensile properties of the TRC system were evaluated by a direct tensile test with a dumbbell-type grip method. Six full-scale slab specimens were fabricated and tested to evaluate the flexural behavior of the RC slabs strengthened with the TRC system. The load-carrying capacity of the slab specimens was evaluated using a four-point load flexural test. The ultimate load-carrying capacity of the strengthened slabs was compared with that of an unstrengthened slab, as well as the analytical solutions.

## 2. Experimental Program

### 2.1. Materials of TRC

The most commonly used textile reinforcement type in the flexural strengthening of concrete structures is two-dimensional grid-type textile [3,4,5]. In this study, a warp-knitted carbon textile (SITgrid 041 KK, Wilhelm Kneitz, Germany) was used as a textile reinforcement (Figure 1a). Table 1 summarizes the material properties of the yarn that was used for the warp and weft direction of the grid. As shown in Figure 1b, the surface of an as-delivered textile was further coated with a fine powder (diameter 69 μm) of aluminum oxide (Al_2_O_3_) and vinyl ester resin to increase the bond strength.

Table 2 provides the mixture composition of mortar used in the TRC system. The motivations, the mix design, and the results of materials tests for the mortar designed in the preceding study are presented in [25].

In this study, polyvinyl alcohol (PVA) fibers (KURALON K-II REC100L, Kuraray, Japan) (Table 3) were incorporated to mitigate the formation of shrinkage-induced cracks. The PVA fibers inhibit the progression of cracks due to a bridging effect, thereby, improving the tensile and flexural strength of the TRC system. The effect of PVA fibers on the mechanical properties of the TRC system was investigated for three different fiber volume fractions (0%, 1%, and 1.5%). The preliminary batch test indicated that if the fiber volume fraction was more than 1.5%, the matrix became too dense to install the TRC system, and hence 1.5% was chosen as the maximum value of the fiber volume fraction.

The compressive strength of air-cured mortar at the time of the test was identified through five cubic mold specimen tests [26]. The mean test values of specimens with 0%, 1.0%, and 1.5% PVA fibers were 70.5, 66.6, and 62.5 MPa, respectively.

### 2.2. Direct Tensile Tests

Many research groups [27,28,29,30] have conducted direct tensile tests to identify the tensile properties of TRC. In their tests, different types of grip method have been used as follows: pin-type [27,28], clevis-type [15], clamped-type [29], and dumbbell-type [19,30]. The preliminary tests indicated that the test with the dumbbell-type grip gave more consistent test results than the clevis-type grip method. In this study, therefore, the tensile properties of the TRC system were evaluated by a direct tensile test with the dumbbell-type grip method (Figure 2a). Figure 2b shows the dimensions of a dumbbell-type specimen. Two 2 mm wide and 12.5 mm deep notches were placed at both middle sides of the coupons to induce a crack at the middle. Vertical monotonic loading with a displacement control of 0.4 mm/min was applied to the specimens using a 300 kN capacity universal testing machine (UTM, Shimadzu, Kyoto, Japan).

Table 4 summarizes the results of the direct tensile tests for the TRC coupons. The influence of the surface coating on the ultimate tensile strength ($f_{fu}$) of the TRC system was insignificant. However, the elastic tensile modulus ($E_{f}$) of the specimens with the coated textile (C-L1 series) was at least 145% greater than that with the uncoated textile (UC-L1 series). This enhancement in $E_{f}$ was obtained because the bonding strength of textile within the matrix was increased by the surface coating. Overall, $f_{fu}$ and $E_{f}$ of the specimens increased as the volume percent of PVA fibers used in the matrix increased. Moreover, $f_{fu}$ and $E_{f}$ of the C-L1 series specimens with 1.5% volume of PVA fibers were increased by 18.3% and 9.6%, respectively as compared with the UC-L1 series specimens. Note that similar behavior was observed for the TRC system with the short dispersed PVA fibers in tension tests [18].

Figure 3 shows the axial stress versus strain curves of the specimens. The strain was calculated by the average value of displacements measured by two laser sensors mounted at both sides of the specimens (Figure 2a). The stress was the applied load divided by the cross-sectional area of the textile (three yarns) within the matrix. In the early loading stage, regardless of the specimen series, the induced stress increased linearly until a matrix crack occurred; however, when the first matrix crack occurred, the magnitude of the sudden stress drop of the uncoated specimens (UC-L1 series) was at least two times greater than that for the coated specimens (C-L1 series). The sudden stress drop of the TRC system after matrix cracking is due to local slip between the textile and matrix. In this study, the sudden stress drop after matrix cracking was reduced for the coated specimens because the surface treatment of the textile improved the bond between the textile and the matrix.

The stress dropped after the first crack, and then the slope decreased but the induced stress continued to increase until the subsequent cracks occurred. It should be noted that the specimens with the PVA fibers showed fewer stress drops than the specimens without the PVA fibers. The stress-strain diagram showed strain softening behavior after the induced stress reached its peak value. The test results indicated that the presence of PVA fibers in the matrix provided crack-bridging force that resisted crack formation. The failure mode of all tested coupon specimens was slippage of textile within the matrix. In this study, none of the specimens showed a failure mode associated with rupture of the textile reinforcement.

### 2.3. Failure Test of Full-Scale Slab Specimens

Seven 1000 × 200 × 2000 mm^3^ (width × height × length) RC slab specimens, i.e., six RC slabs strengthened with the TRC system and one unstrengthened RC slab (considered as the control specimen), were fabricated for a full-scale flexural test (Table 5). Figure 4a illustrates the reinforcement details where a 10 mm bar is uniformly spaced at 165 mm in the transverse direction of the slab specimen. The mean value of yield strength of a 16 mm bar and the compressive strength of the air-cured normal density concrete were 477 and 30.2 MPa, respectively.

Figure 4b shows a strengthening plan for the RC slab specimen with the TRC system. For convenience in applying the TRC system over the slab specimens, the slab specimens were casted in an inverted position. As shown in Figure 5a, the top surface of the slabs was ground to roughly 2–3 mm in depth with a power tool after a seven day curing period. The fresh mortar was poured onto the steel formwork to form the 1st layer of the TRC system. The surface of the 1st layer of the TRC system was finished smoothly using a trowel (Figure 5b), and then the carbon textile was placed onto the mortar surface (Figure 5c). Finally, the fresh mortar was poured onto the textile, and then the top surface of the TRC system was finished (Figure 5d). Figure 6 shows the configuration of the TRC system with two different thicknesses.

Figure 7a shows a typical set-up and instrumentation (LVDT and strain gauges) for a four-point flexural test. The flexural test was conducted using a 2000 kN capacity UTM (Daekyung, Seoul, Korea) with displacement control at a speed of 1.2 mm/min (Figure 7b).

## 3. Test Results and Discussion

### 3.1. Load-Carrying Capacity

Figure 8 shows the load versus mid-span vertical displacement curves of the RC, S15-1, and S20-1 specimens. Overall, the TRC strengthened slab specimens as well as the unstrengthened slab specimen showed approximately tri-linear load-displacement behavior that reflects the following three distinct phases: concrete cracking, tension steel bar yielding, and failure. The initial concrete cracking load, steel rebar yield load, and ultimate load of the strengthened specimen S15-1 were at least 109%, 139%, and 165% greater than those of the unstrengthened specimen, respectively. Although the concrete cracking loads of the S15-1 and S20-1 specimens were greater than that of the RC specimen, the stiffness of all specimens was almost identical, up to the point of concrete cracking.

After the concrete cracking, the stiffness of the strengthened specimens was slightly decreased but the load-displacement curve was still linear until the steel rebar yield. The stiffness of the unstrengthened specimen dramatically decreased after the steel yield. However, after the steel yielding, the load of the strengthened specimens was continuously increased until failure. The failure load of the strengthened specimens S15-1 and S20-1 was at least 134%, 125% greater than that of the steel yield load, respectively. Furthermore, the stiffness values of the S15-1 and S20-1 specimens, defined as the slope between concrete cracking and the steel yielding phase, were 112% and 123% of the unstrengthened specimen, respectively. Therefore, the test results indicated that the TRC system was effective in increasing the flexural capacity, as well as the flexural stiffness of RC slab elements.

Overall, the flexural capacity of the strengthened specimens increased as the number of textile plies was increased, but the increase was not linearly proportional to the amount of textile reinforcement. Note that similar behavior was also observed for RC flexural members strengthened with different numbers of textile plies in flexural tests [8,10,11]. The failure test results indicated that the ultimate load carrying capacity of the S20-1 specimen (two-ply textile) was increased by 11% only relative to that of the S15-1 specimen (one-ply textile). The strengthening effect was underestimated because the failure mode of the S20 series specimens changed from a flexure to a flexure-shear mode as the number of plies of the textile was increased.

Figure 9a shows the load versus mid-span vertical displacement curves of the S15 series specimens. The initial cracking load and steel yield load of the S15 series specimens slightly increased as the volume of short fibers increased. However, the influence of short fibers on the concrete cracking and ultimate load of the strengthened specimens was not significant. Figure 9b shows the load versus mid-span vertical displacement curves of the S20 series specimens. The ultimate load-carrying capacity of the S20-1 and S20-2 specimens was at most 88% of the S20-3 specimen. It should be noted that the S20-1 and S20-3 specimens failed in flexure-shear mode whereas the S20-2 specimen failed in shear mode due to the TRC strengthening effect. The shear failure load level of the slab specimen estimated using the nominal shear capacity was 235 kN. Although the test data for the S20-2 specimen were presented, these data were not compared with those of the S20-1 and S20-3 specimens because the S20-2 specimen showed a premature shear failure.

Therefore, the enhancement of the flexural capacity with the TRC system should carefully be calculated in the strengthening design process to avoid an undesirable failure mode. Otherwise, depending on the shear span ratio and the level of strengthening, strengthening of flexural members with the TRC system can lead to a change of the failure mode from flexural to shear failure mode or flexure-shear mode. This is because the TRC system acts as a tensile reinforcement in RC flexural elements, and hence the TRC flexural strengthening increases the tensile reinforcement ratio.

### 3.2. Crack Patterns and Failure Mode

Figure 10 shows the crack patterns of the specimens after failure, whereas Figure 11 illustrates the progress of cracks, marked with the applied load level. The RC specimen (control) showed a smaller number of typical flexural cracks but larger crack width as compared with the TRC strengthened slab specimens (S15 and S20 series). The TRC flexural strengthening, therefore, was an effective means of forming fine cracks while inhibiting crack opening. The RC specimen showed a pure flexural failure mode, whereas the TRC strengthened slab specimens showed a flexure-shear failure mode.

The TRC strengthened slab specimens (S15 and S20 series) showed mainly flexural cracks until the load level reached the steel rebar yield load. However, a number of diagonal shear cracks spread toward the top of the concrete when the applied load was further increased. The diagonal shear cracks were more profound for the S20 series specimens that were strengthened with two-ply textile than the S15 series specimens. Generally, the crack pattern of the RC flexural members also changes from full flexure to a combination of flexure and shear as the tensile reinforcement ratio increases [31].

For the S20-1 and S20-2 specimens, fewer through-thickness cracks of the TRC system were observed than in the S20-3 specimen. The load level that caused a diagonal shear failure of the S20-1 and S20-2 specimens was 335 kN and 250 kN, respectively. Therefore, the ultimate behavior of the S20-1 and S20-2 specimens was affected by the shear failure mode rather than the flexure failure mode.

Figure 12 illustrates a fractured yarn of the textile reinforcement within the cracked section of the S15-1 specimen after failure. In general, the results of the experimental investigations by other research groups indicated that the flexural members strengthened with one-ply textile failed by slippage of textile within the matrix. Additionally, delamination of the TRC system from the concrete substrate occurred for the flexural members strengthened with four-ply textile [8,11]. However, no delamination of the TRC system from the concrete substrate or slip of the textile reinforcement within the matrix were detected for the S15 and S20 series specimens during the flexural test. This apparent beneficial behavior was obtained because good bonding between the textile and matrix was obtained in this study by the surface coating of the textile.

### 3.3. Analytical Solutions

An analytical method to compute the flexural capacity of RC elements strengthened with the TRC system was suggested by the ACI Committee 549 [2]. In this study, the ultimate load-carrying capacity of the slab specimen was estimated by the design method in ACI 549.4R-13 [2] and analytical procedures presented in [25].

The ultimate load-carrying capacity ($P_{u}$) of the slab specimen strengthened with the TRC system can be computed according to Equation (1).
(1)$$P_{u}=\frac{2M_{u}}{l_{p}}$$
where, $M_{u}$ is the ultimate flexural moment and $l_{p}$ the distance from the support to the location of the point load (Figure 7a). In Equation (1), $M_{u}$ is computed according to Equation (2)
(2)$$M_{u}=M_{S}+M_{f}$$
where, $M_{S}$ and $M_{f}$ are the moment contribution of steel rebars and textile to flexural capacity, respectively.

In Table 6, the ultimate loads obtained in the test for the slab specimens ($P_{u}^{Test}$) were compared with the analytical calculations. The tested values of yield strength of steel (477 MPa) and compressive strength of concrete (30.2 MPa) were used in the analytical calculations. In the table, two analytical calculations are provided, where $P_{u}^{A}$ used $E_{f}$ were obtained in the direct tensile test conducted in this study (Table 4), and $P_{u}^{B}$ utilized $E_{f}$ of a bare textile (Table 1).

$P_{u}^{A}$ and $P_{u}^{B}$ were approximately 73% and 91% of the test data, respectively, and notably, $P_{u}^{A}$ was at most 79% of $P_{u}^{B}$. This is due to the inclusion of the slippage of the textile within the matrix in the calculation of $P_{u}^{A}$. Note that the direct tensile test conducted in this study used a dumbbell-type grip method where all specimens failed by slippage of textile within the matrix. However, Arboleda et al. [32] observed that the value of $E_{f}$ obtained in the direct tensile test of the TRC system with a clamping grip is similar to that of a bare textile. Therefore, if the textile embedded in the matrix has a sufficient development length, then, $E_{f}$ of a bare textile can be used in the analytical calculation.

### 3.4. Design and Application Consideration

As presented in Section 3.1, the load-displacement curve of the unstrengthened slab specimen showed ductile behavior after steel yielding. Although the load capacity and the stiffness of the slab specimens were significantly increased by the flexural strengthening with the TRC system, the strengthened specimens showed a sudden load drop after failure. It is apparent that an unyielding characteristic in tension of carbon fibers caused brittle failure of the TRC strengthened members. Moreover, the failure of the RC flexural members strengthened with the TRC system is often associated with delamination of the TRC system from the concrete substrate or bonding failure of the textile reinforcement. The brittle failure of the TRC system must be considered in the design process of flexural members strengthened with the TRC system to avoid a catastrophic failure.

In Table 7, roughly estimated material, labor, and the total cost of the TRC flexural strengthening work for RC slab are compared with those of epoxy-bonded carbon sheet flexural strengthening work. Note that the TRC system (15 mm thick, 0% PVA fibers) was assumed to be installed by a shotcrete machine. The estimated total cost of the TRC strengthening system was 57% of that of the epoxy-bonded carbon sheet strengthening system. Other advantageous features of the TRC strengthening system over the epoxy-bonded carbon sheet strengthening system are superior fire resistance and the fact that the TRC system can be applied to a wet concrete surface [33].

In this study, spacers to fix the grid reinforcement were not used during the installation of the TRC system since the slab specimens were casted in an inverted position. However, if the TRC system is to be installed underneath the RC slab elements or curved RC elements, an efficient spacer or fixing method for the textile should be developed for practical application.

## 4. Conclusions

The flexural strengthening of RC slab-type elements with a carbon TRC system was experimentally studied and presented in this paper. The influence of surface coating of the incorporated textile, the short fiber volume fractions, and number of textile plies of the TRC system on the flexural capacity of RC slabs were investigated. The results of the flexural failure test indicated that the stiffness and the ultimate flexural capacity of the strengthened slab was at least 112% and 165% greater than that of the unstrengthened slab, respectively. Therefore, if properly designed and installed, the TRC system can effectively be used for flexural strengthening of the RC slab elements.

The influence of the surface coating of the textile on the bonding behavior of the textile within the TRC system was significant. The ultimate strain of the TRC system with the surface-coated textile was 69% of that with the uncoated textile. Therefore, the surface coating of the textile was effective in increasing the elastic tensile modulus of the TRC system.

The matrix was reinforced with short fibers to mitigate shrinkage-induced crack formation. The PVA fibers were very effective to mitigate shrinkage-induced crack formation during the installation of the TRC system to the RC slab specimen. Although incorporating the PVA fibers within the matrix enhanced the tensile properties of the TRC system, the PVA fibers did not have a significant influence on the concrete cracking and ultimate load of the full-scale slab specimens. This could be due to the fact that the concrete cracking load level of the full-scale slab specimen is many times greater than the matrix cracking load (stress) level of the TRC tensile coupon specimen.

The flexural capacity of the strengthened specimens increased as the number of textile plies in the TRC system increased. However, the strengthening effect was not linearly proportional to the amount of textile reinforcement. Furthermore, the failure test for the slab specimens indicated that the TRC flexural strengthening with an increased number of textile plies affected the failure mode and also caused shear failure of the flexural members.

## Figures and Tables

**Figure 1 materials-13-02246-f001:**
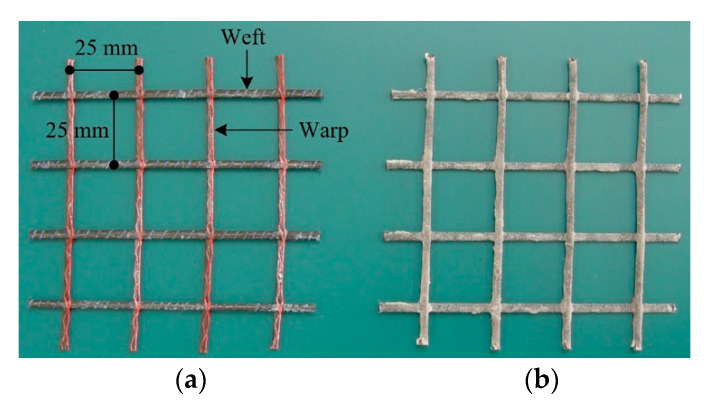
Carbon textile grid. (**a**) As-delivered condition; (**b**) Surface coated by Al_2_O_3_ powder.

**Figure 2 materials-13-02246-f002:**
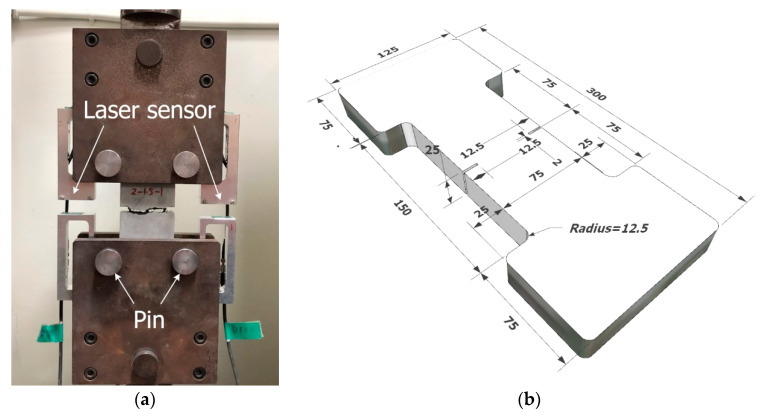
Schematic of direct tensile test. (**a**) Test set-up; (**b**) Dimensions of coupon specimen (units, mm).

**Figure 3 materials-13-02246-f003:**
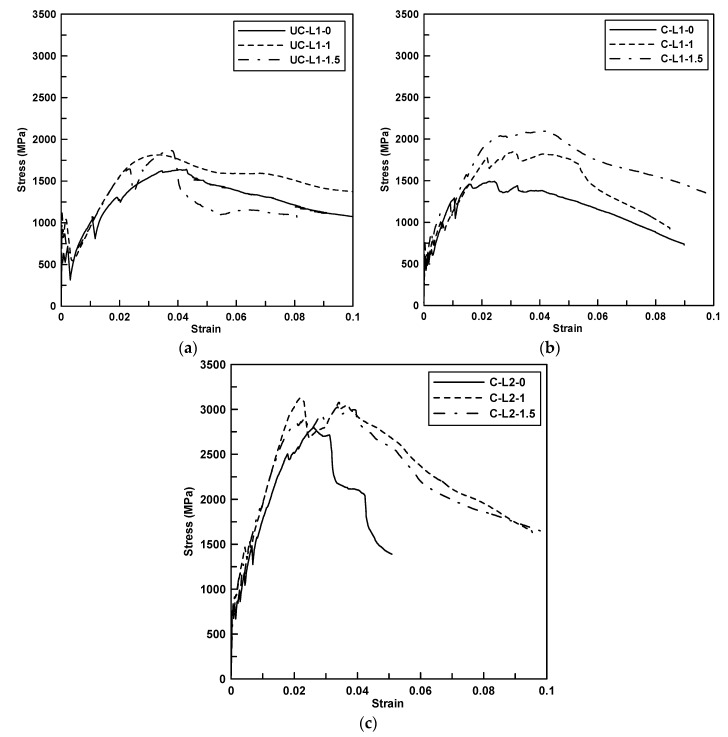
Axial stress versus strain curve of direct tension tested specimens. (**a**) UC-L1 series; (**b**) C-L1 series; (**c**) C-L2 series.

**Figure 4 materials-13-02246-f004:**
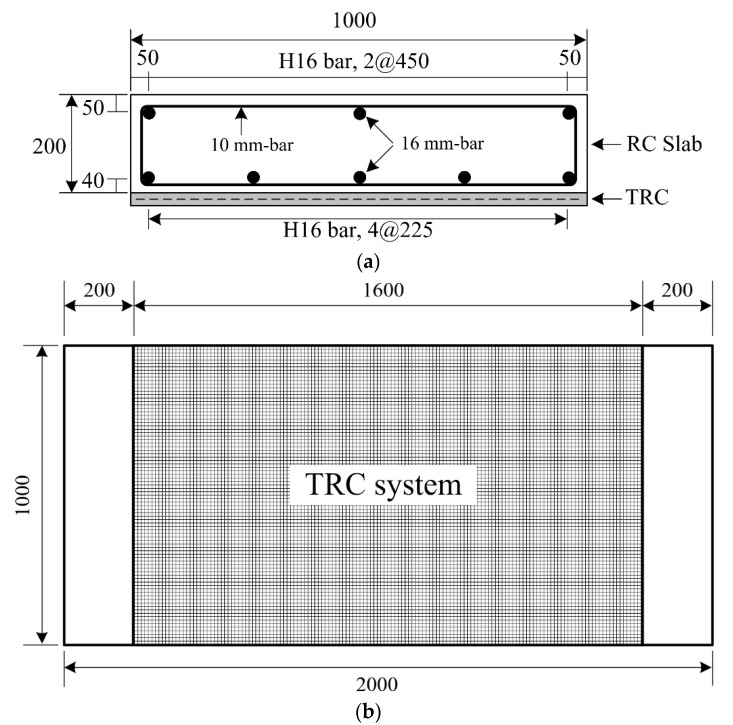
Dimensions of a full-scale slab specimen. (**a**) Cross-sectional view; (**b**) Plan view (units: mm).

**Figure 5 materials-13-02246-f005:**
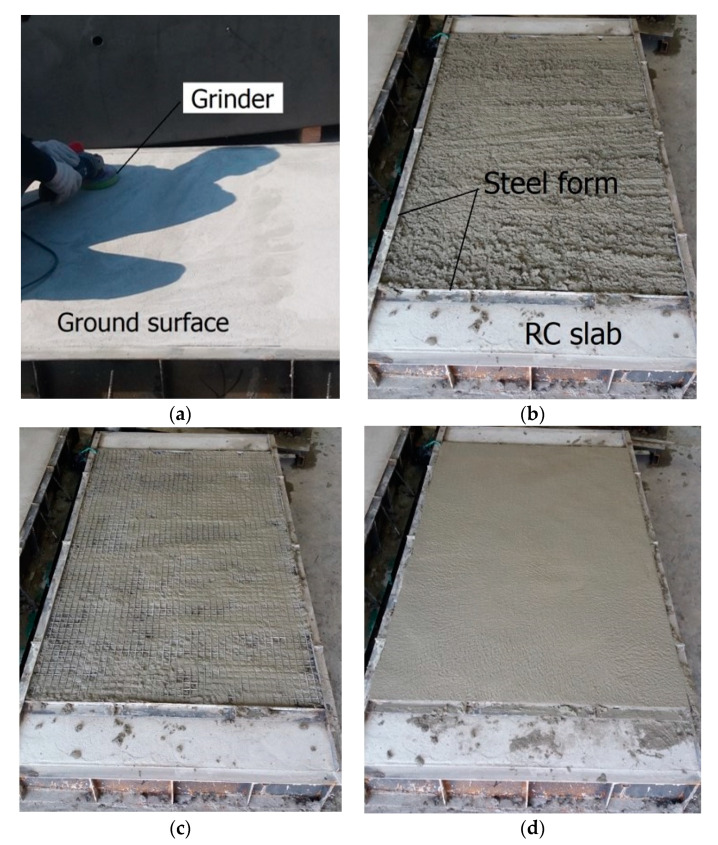
Fabrication process of the slab specimen. (**a**) Grinding of top surface; (**b**) Placing of 1st mortar layer; (**c**) Placing textile reinforcement; and (**d**) Placing of 2nd mortar layer.

**Figure 6 materials-13-02246-f006:**
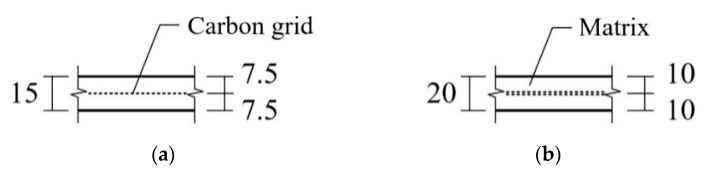
Configuration of the textile reinforced concrete (TRC) system. (**a**) 15 mm thick (one-ply textile); (**b**) 20 mm thick (two-ply textile) (units, mm).

**Figure 7 materials-13-02246-f007:**
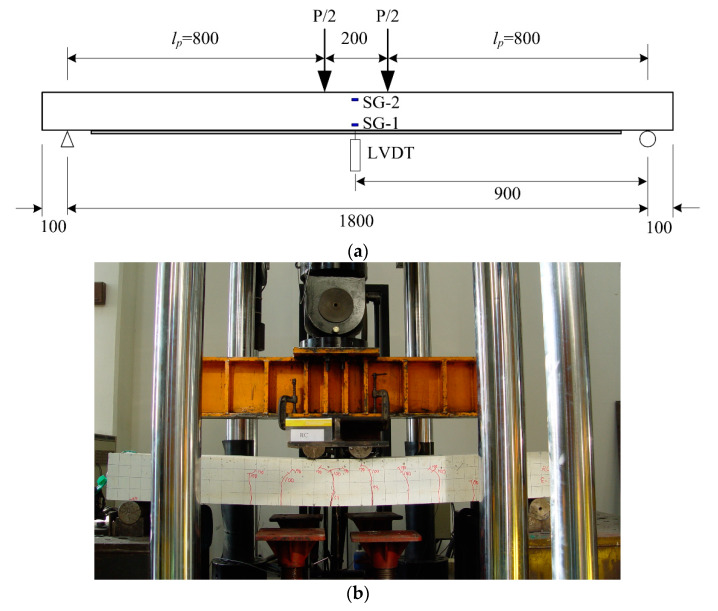
Full-scale flexural test setup and instrumentation. (**a**) Layout; (**b**) Photo (units, mm).

**Figure 8 materials-13-02246-f008:**
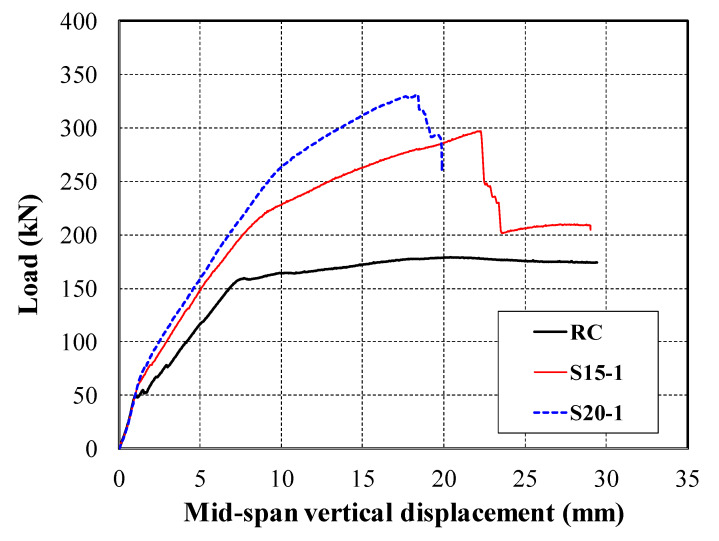
Comparison of load versus mid-span displacement curves of reinforced concrete (RC), S15-1, and S20-1 specimens.

**Figure 9 materials-13-02246-f009:**
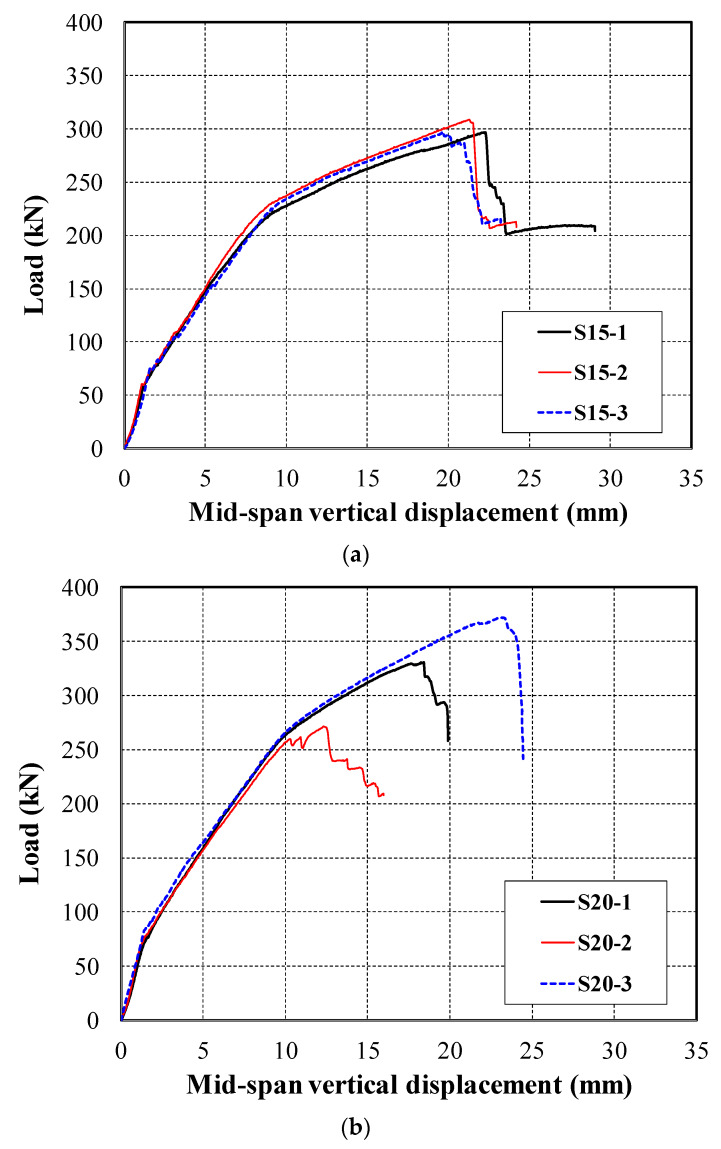
Load versus mid-span displacement curves of the specimens. (**a**) S15 series; (**b**) S20 series.

**Figure 10 materials-13-02246-f010:**
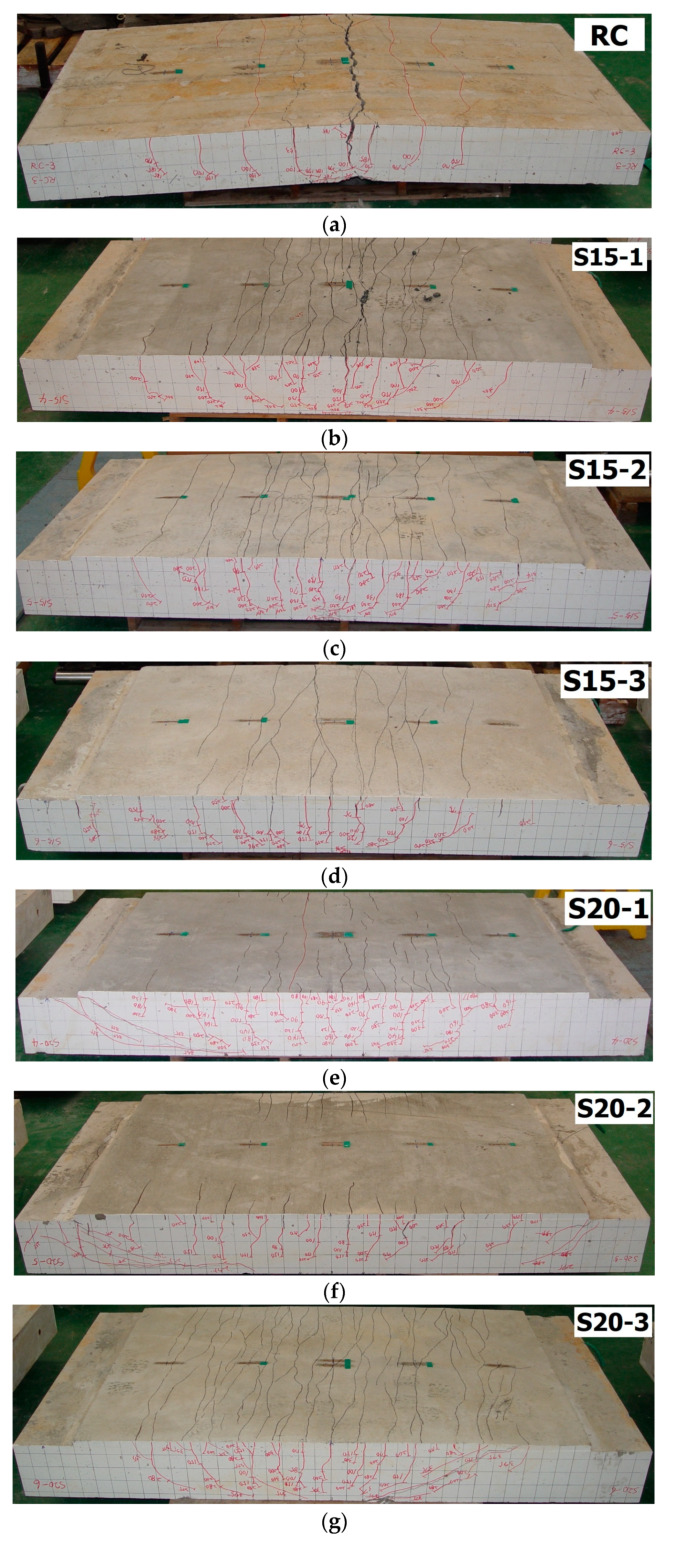
Cracked patterns on the bottom side of specimens after failure (photo). (**a**) RC; (**b**) S15-1; (**c**) S15-2; (**d**) S15-3; (**e**) S20-1; (**f**) S20-2; (**g**) S20-3.

**Figure 11 materials-13-02246-f011:**
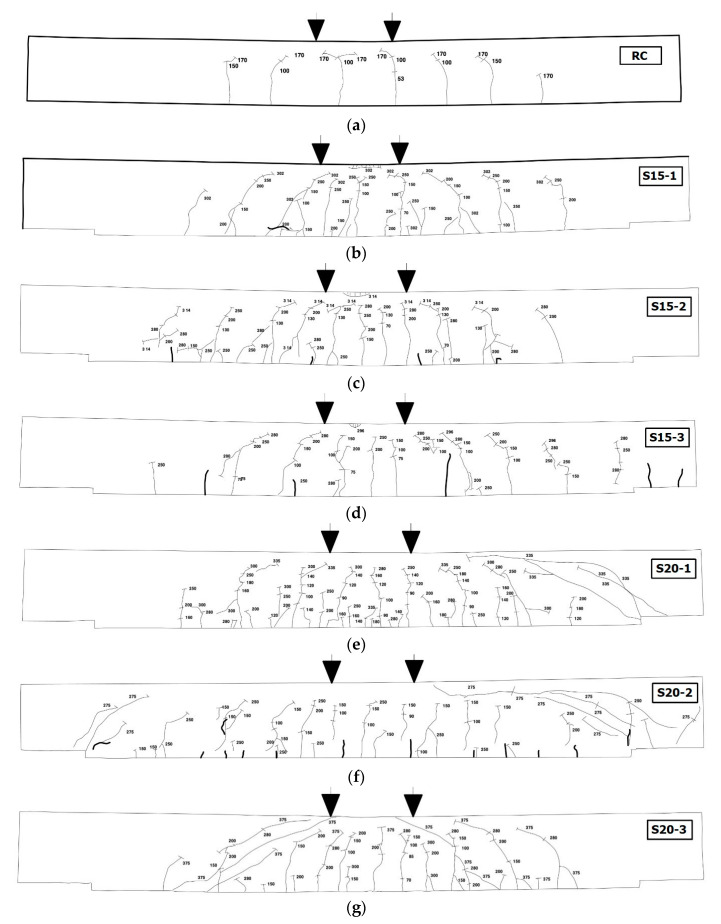
Progress of cracks on the side of specimens marked with the applied load level. (**a**) RC; (**b**) S15-1; (**c**) S15-2; (**d**) S15-3; (**e**) S20-1; (**f**) S20-2; (**g**) S20-3.

**Figure 12 materials-13-02246-f012:**
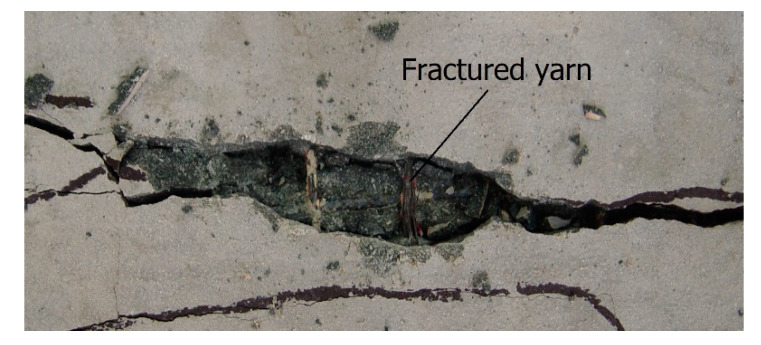
Fracture of textile at the mid-span of S15-1 specimen.

**Table 1 materials-13-02246-t001:** Material properties of carbon grid-type textile in warp direction (suggested values by the manufacturer).

Fiber	Resin	Cross-Sectional Area of Yarn (mm^2^)	Tensile Strength (MPa)	Elastic Modulus (GPa)	Elongation (%)
3200 tex	Polystyrene	1.808	1700	200	1.0

**Table 2 materials-13-02246-t002:** Mixture composition of binder (unit, kg/m^3^).

Cement	Granulated Blast-Furnace Slag	Sand	Water	Superplasticizer
466	466	1024	278	3

**Table 3 materials-13-02246-t003:** Properties of polyvinyl alcohol (PVA) fiber (suggested values by the manufacturer).

Length (mm)	Diameter (dtex)	Modulus of Elasticity (cN/dtex)	Tenacity (cN/dtex)	Density (g/mm^3^)	Elongation (%)
6.12	100	220	10	1.3	13

**Table 4 materials-13-02246-t004:** Summary of the results for direct tension tests (average values of ten tests).

Specimen ID	Surface Coating	No. of Textile Layer	PVA Fiber (vol.%)	Thickness (mm)	$f_{fu}(\mathrm{MPa})$	$E_{f}(\mathrm{MPa})$	CoV
UC-L1-0	Uncoated	1	0	15	1488	40,242	0.130
UC-L1-1	Uncoated	1	1	15	1639	47,100	0.158
UC-L1-1.5	Uncoated	1	1.5	15	1688	40,541	0.167
C-L1-0	Coated	1	0	15	1542	58,276	0.083
C-L1-1	Coated	1	1	15	1661	61,717	0.073
C-L1-1.5	Coated	1	1.5	15	1824	63,843	0.057
C-L2-0	Coated	2	0	20	2709	95,676	0.111
C-L2-1	Coated	2	1	20	2809	97,689	0.108
C-L2-1.5	Coated	2	1.5	20	3000	98,618	0.114

**Table 5 materials-13-02246-t005:** Dimensions of full-scale slab specimens.

Specimen ID	TRC Thickness (mm)	No. of Textile Ply	PVA Fiber (vol.%)	Slab Thickness (mm)
RC	−	−	−	200
S15-1	15	1	0	215
S15-2	15	1	1.0	215
S15-3	15	1	1.5	215
S20-1	20	2	0	220
S20-2	20	2	1.0	220
S20-3	20	2	1.5	220

**Table 6 materials-13-02246-t006:** Summary of the test and analysis results for the slab specimens.

Specimen ID	Experiment		Analytical Calculations		Comparison	
	$P_{u}^{Test}(\mathrm{kN})$	Failure Mode	$P_{u}^{A}(\mathrm{kN})$	$P_{u}^{B}(\mathrm{kN})$	PuAPuTest	PuBPuTest
RC	179.6	Flexure	184.5	184.5	1.03	1.03
S15-1	296.8	Flexure	212.1	267.8	0.71	0.90
S15-2	309.1	Flexure	213.7	267.8	0.69	0.87
S15-3	297.3	Flexure	214.6	267.8	0.72	0.90
S20-1	330.4	Flexure/shear	267.0	332.2	0.81	1.01
S20-2	276.6	Shear	−	−	−	−
S20-3	371.9	Flexure/shear	269.1	332.2	0.72	0.89

**Table 7 materials-13-02246-t007:** Summary of the material, labor, and the total cost of the flexural strengthening (cost units, US$/m^2^).

Method	No. of Ply	Material Cost ($)	Labor Cost ($)	Total Cost ($)
TRC system	1	52	40	93
Epoxy-bonded carbon sheet	1	75	90	165

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
