# Peer review of "Flexural Strengthening of Concrete Slab-Type Elements with Textile Reinforced Concrete"

_materials, 2020, doi:10.3390/ma13102246_

Round 1

Reviewer 1 Report

Congratulations to authors. Very interesting work regarding a valuable material used in concrete: Carbon fibers.

In general the paper is well presented but some general information may be clarified or completed:

  1. The “Abstract” should be more than a repetition of some paragraphs extracted from the text;
  2. Lines 64/65: does the “warp knitted carbon textile” have the same characteristics in both directions? “Figure 1. (a)” shows that there is a different material in the other direction;
  3. Line 87: can the Authors better explain all motivations that led to the choice of the mixture composition referred to in Table 2? In lines 77/79 only the use of a “large amount of a granulated blast-furnace slag (GGBS)” is explained;
  4. Tables 1 & 3: can Authors include references to the standards used?
  5. Line 131: please change “Seven 1000 x 200 x 2000 mm (width …” by “Seven 1000 x 200 x 2000 mm3 (width …” or (worst) “Seven 1000 mm x 200 mm x 2000 mm (width …”;
  6. Lines 191, 222 & 224: please change “Mid-span vertical diaplacement (mm)” by “Mid-span vertical displacement (mm)”;
  7. The authors may include some information regarding one important aspect for the application industry: costs of each solution. Please, include for example a table with a percentage comparison between average direct costs.

Author Response

  1. The “Abstract” should be more than a repetition of some paragraphs extracted from the text;

Authors’ Response and Revision: Lines 9- 23: Abstract has completely been rewritten in the revised manuscript.

  1. Lines 71: does the “warp knitted carbon textile” have the same characteristics in both directions? “Figure 1. (a)” shows that there is a different material in the other direction;

Authors’ Response and Revision: Yes. An additional explanation has been added. Lines 72-74: Table 1 summarizes the material properties of the yarn that used for the warp and weft direction of the grid; Labels for the warp and weft have been added to Figure 1.

  1. Line 89: can the Authors better explain all motivations that led to the choice of the mixture composition referred to in Table 2? In lines 86/88 only the use of a “large amount of a granulated blast-furnace slag (GGBS)” is explained;

Authors’ Response and Revision: The objective, mix design procedures, and detailed results of material tests for the mortar have been presented in the preceding paper by the authors. To avoid repetition of the results of the preceding study, the sentence has been changed as: Lines 90-96; Table 2 provides the mixture composition of mortar used in the TRC system. The motivations, the mix design, and the results of materials tests for the mortar designed in the preceding study are presented in [25].

  1. Tables 1 & 3: can Authors include references to the standards used?

Authors’ Response and Revision: The material properties of the textile and PVA fiber in Tables 1 and 3 are the suggested values by the manufacturers. Line 80-81: Table 1. Material properties of carbon grid-type textile in warp direction (suggested values by the manufacturer); Line 100: Table 3. Properties of PVA fiber (suggested values by the manufacturer).

  1. Line 147: please change “Seven 1000 x 200 x 2000 mm (width …” by “Seven 1000 x 200 x 2000 mm3 (width …” or (worst) “Seven 1000 mm x 200 mm x 2000 mm (width …”;

Authors’ Response and Revision: Changed. Line 147; The sentence has been changed as “Seven 1000 mm x 200 mm x 2000 mm (width …”;

  1. Lines 200, 235 & 237: please change “Mid-span vertical diaplacement (mm)” by “Mid-span vertical displacement (mm)”;

Authors’ Response and Revision: Corrected. Lines 200, 235 & 237; Syntax error have been corrected.

  1. The authors may include some information regarding one important aspect for the application industry: costs of each solution. Please, include for example a table with a percentage comparison between average direct costs.

Authors’ Response and Revision: Lines 323-329: Roughly estimated material, labor, and the total cost of the TRC flexural strengthening work for RC slab are compared with those of epoxy-bonded carbon sheet flexural strengthening work.

Reviewer 2 Report

  • The article still needs some grammatical and syntax improvements. Use of English service center is recommended.
  • The introduction needs to be revised for higher quality language. Using separated paragraphs is encouraged but with more details. In addition, the authors mentioned some works without stating about the contributions, pros and cons and the how the current work would address.
  • The authors mentioned about the “strengthening of concrete structures” and “Early developments of TRC and recent applications to the flexural strengthening of
    30 concrete structures are well summarized in Refs”, the following more recent works are recommend to be considered
    • Farzampour, A. (2019). Compressive behavior of concrete under environmental effects. IntechOpen.
    • Farzampour, A. (2017). Temperature and humidity effects on behavior of grouts. Advances in concrete construction, 5(6), 659.
  • In addition, recent works related to “short fibers to mitigate shrinkage-induced crack formation” and “studied the influence of short fibers on the mechanical properties of TRC” on the fibers effect in the concrete is suggested to be considered as well.
    • Mansouri, I., Sadat Shahheidari, F., Hashemi, SMA., & Farzampour, A. (2020) Investigation of steel fiber effects on concrete abrasion resistance, Advances in concrete construction, 9(4), 367-374.
  • The concrete mixing design method should be defined with more details.
  • Figure 7.a, is the 1600 correct?
  • What is reason for change of mode of behavior if the Slab is over strengthened with TRC system?
  • What is the reason for less cracks for S20-2 compared to S20-1 and S20, 3 based on the Figure 10?
  • S 20 crack conditions should be compared with S15 ones as well.
  • Why the fibers are considered in 0,% 1%, and 1.5%. The work below could be implemented for comparison:
    • Mansouri, I., Sadat Shahheidari, F., Hashemi, SMA., & Farzampour, A. (2020) Investigation of steel fiber effects on concrete abrasion resistance, Advances in concrete construction, 9(4), 367-374.
  • The Figures should be more detailed. For example, Figure 1 needs to be more detailed. The type of bars, size of bars, etc.
  • What is the reason for “the magnitude of sudden stress drop for the uncoated specimens (UC-L1 series) was at least two times greater than that for the coated specimens”
  • Equations in part 3.3 should be included and referenced. Also, what is the reason of more than 30% difference in the values from the specimen and the analytical equations in some cases
  • The application of the study and the shortcomings are recommended to be included.
  • It is mentioned that the PVA fiber did have a significant influence on the concrete cracking. What would be the reason for such behavior as opposed to several studies showing that the fibers could useful for crack control

Author Response

  • S 20 crack conditions should be compared with S15 ones as well.

Authors’ Response and Revision: Lines 269-271: The diagonal shear cracks were more profound for the S20 series specimens that were strengthened with two-ply textile than the S15 series specimens.

  • Why the fibers are considered in 0,% 1%, and 1.5%. The work below could be implemented for comparison:

Mansouri, I., Sadat Shahheidari, F., Hashemi, SMA., & Farzampour, A. (2020) Investigation of steel fiber effects on concrete abrasion resistance, Advances in concrete construction, 9(4), 367-374.

Authors’ Response and Revision: The above paper is related to the effects of steel fibers on concrete abrasion resistance. Thus, the authors believe that the results of the above paper are not suitable for comparison purpose. On the other hand, the reason for choosing the fiber volume fraction has been added. Lines 94-96: The preliminary batch test indicated that if the fiber volume fraction was more than 1.5%, the matrix became too dense to install the TRC system, and hence 1.5% was chosen as the maximum value of the fiber volume fraction.

  • The Figures should be more detailed. For example, Figure 1 needs to be more detailed. The type of bars, size of bars, etc.

Authors’ Response and Revision: Labels to describe the details have been added in Figure 1 and Figure 4.

  • What is the reason for “the magnitude of sudden stress drop for the uncoated specimens (UC-L1 series) was at least two times greater than that for the coated specimens”

Authors’ Response and Revision: Lines 128-131: The sudden stress drop of the TRC system after matrix cracking is due to local slip between the textile and matrix. In this study, the sudden stress drop after matrix cracking was reduced for the coated specimens because the surface treatment of the textile improved the bond between the textile and the matrix.

  • Equations in part 3.3 should be included and referenced. Also, what is the reason of more than 30% difference in the values from the specimen and the analytical equations in some cases

Authors’ Response and Revision: Lines 293-299: Eq. (1) and Eq. (2) have been added; Lines 305-307: Note that  PAu  was at most 79% of PBu . This is due to the fact that inclusion of the slippage of the textile within the matrix in the calculation of PAu.

  • The application of the study and the shortcomings are recommended to be included.

Authors’ Response and Revision: Lines 323-330: In Table 7, roughly estimated material, labor, and the total cost of the TRC flexural strengthening work for RC slab are compared with those of epoxy-bonded carbon sheet flexural strengthening work. Note that the TRC system (10 mm-thick, 0% PVA fibers) was assumed to be installed by a shotcrete machine. The estimated total cost of the TRC strengthening system was 57% of that of the epoxy-bonded carbon sheet strengthening system. Other advantageous features of the TRC strengthening system over the epoxy-bonded carbon sheet strengthening system are superior fire resistance and that the TRC system can be applied to a wet concrete surface; In this study, spacers to fix the grid reinforcement were not used during the installation of the TRC system since the slab specimens were casted in an inverted position. However, if the TRC system is to be installed underneath RC slab elements or curved RC elements, an efficient spacer or fixing method for the textile should be developed for practical application.

  • It is mentioned that the PVA fiber did have a significant influence on the concrete cracking. What would be the reason for such behavior as opposed to several studies showing that the fibers could useful for crack control.

Authors’ Response and Revision: Lines 347-353: The matrix was reinforced with short fibers to mitigate shrinkage-induced crack formation. The PVA fibers were very effective to mitigate shrinkage-induced crack formation during the installation of the TRC system to the RC slab specimen. Although incorporating the PVA fibers within the matrix enhanced the tensile properties of the TRC system, the PVA fibers did not have a significant influence on the concrete cracking and ultimate load of the full-scale slab specimens. This may be due to the fact that the concrete cracking load level of the full-scale slab specimen is many times greater than the matrix cracking load (stress) level of the TRC tensile coupon specimen.

Round 2

Reviewer 2 Report

The English syntax and grammatical issues are recommended to be checked and revised. 

Author Response

• The English syntax and grammatical issues are recommended to be checked and revised.

Authors’ Response and Revision:

Lines 210-211: The sentence has been changed as “Overall, the flexural capacity of the strengthened specimens increased as the number of textile plies was increased, but the increase was not linearly proportional to the amount of textile reinforcement.”

Line 266: Changed as “Figure 11. Progress of cracks on the side of specimens marked with the applied load level.”

Line 297: Changed “In (1)” to “In Eq. (1)”